# Superiority of MRI for Evaluation of Sacral Insufficiency Fracture

**DOI:** 10.3390/jcm11174968

**Published:** 2022-08-24

**Authors:** Taro Yamauchi, Sagar Sharma, Sarath Chandra, Masato Tanaka, Yoshihiro Fujiwara, Shinya Arataki, Ayush Sharma, Yusuke Yokoyama, Toshinori Oomori, Akihiro Kanamaru, Shin Masuda, Noriyuki Shimizu, Kenta Torigoe, Osamu Honda

**Affiliations:** 1Department of Orthopedic Surgery, Okayama Rosai Hospital, 1-10-25 Chikkomidorimachi, Minami Ward Okayama, Okayama 702-8055, Japan; 2Department of Orthopedic Surgery, Smt. N.H.L. Municipal Medical College, Pritan Rai Cross Road, Ellise Bridge, Paldi, Ahmedabad 380006, India; 3Department of Radiology, Okayama Rosai Hospital, 1-10-25 Chikkomidorimachi, Minami Ward Okayama, Okayama 702-8055, Japan

**Keywords:** sacral insufficiency fracture, MRI, CT, plain radiograph

## Abstract

Study Design: Retrospective observational study. Background: Sacral insufficiency fractures (SIF) are relatively rare fractures and difficult to diagnose on plain radiographs. The primary objective of the present study was to evaluate the role of lumbar magnetic resonance imaging (MRI) for the diagnosis of SIF. The secondary objective was to identify the classification of SIF by computed tomography (CT). Methods: A total of 77 (Male 11, female 66, mean 80.3 years) people were included in this study. Inclusion criteria for this study were: age ≥ 60 years and no history of high energy trauma. Exclusion criteria were high energy trauma and a current history of malignancy. Differences in the fracture detection and description in the various radiologic procedures were evaluated. Fracture patterns were evaluated with CT. The detection rates of additional pathologies in the MRI of the pelvis and lumbar spine were also recorded. Results: The sensitivities for SIF were 28.5% in radiographs and 94.2% in CT, and all fractures were detected in MRI. MRI showed a more complex fracture pattern compared with CT in 65% of the cases. We observed 71.4% of single SIFs, 9.1% with other spinal fractures, 13.0% with other pelvic fractures, and 7.8% with other fractures. According to the SIF fracture pattern, the H/U type was 40.2%, transverse type was 33.7%, λ/T type was 24.7%, unilateral vertical type was 1.3%, and bilateral vertical type was 0%. Conclusions: an MRI of the lumbar spine including the sacrum with a coronal fat-suppressed T2-weighted image is useful for elderly patients with suddenly increasing low back pain at an early stage. This procedure improves an early SIF detection, recognition of concomitant pathologies, and adequate treatment for the patients.

## 1. Introduction

Human bone becomes more prone to fracture and incidences of osteoporotic or insufficient fracture increases with age. Age-related fracture is an increasing health and economic concern and increased 50% in twenty years in the USA [1]. In Asia, more than 50% of worldwide osteoporotic fractures will occur by 2050 [2]. In Europe, the cost of osteoporosis will be 37 billion euros [2]. In order to resize the impact of age-related bone fractures and to start reducing their health and economic influence, early diagnosis is key [3].

Sacral insufficiency fractures (SIF) have been commonly reported in patients with severe osteoporosis [4], long spinal fusion [5], and pelvic irradiation [6]. Typically, patients had no history of trauma [7] and no specific symptoms compared with other lumbar spine pathologies [8]. Diagnosis of SIF is difficult with plan radiographs [9], so lumbar and pelvic MRI are very important in order to avoid missed diagnoses [10].

The goals for the treatment of SIF are pain control, early mobilization, and preventing other morbidities. Delayed diagnosis can lead to complications like depression, dementia, cardiac/pulmonary complications, deep venous thrombosis/pulmonary embolization, and disuse syndrome [11]. Early diagnosis is the key to preventing these complications and helps in adequate treatment [12]. The study was conducted in Japan where the elderly form a major part of the active society and where it varies with racial, social, and pathological considerations [13]. Therefore, the primary objective of this study was to evaluate the sensitivity in SIF according to plain radiographs, CT, and MRI. The second objective was to identify SIF fracture patterns by our classification.

## 2. Materials and Methods

### 2.1. Patient Population

The present study was approved by our institutional review board (No. 332). We obtained fully informed consent from each patient prior to participation in this study. From June 2018 to January 2022, a total of 77 patients with SIF who were admitted at our hospital were included in the present study (Table 1). A diagnosis for SIF was made and confirmed by clinical symptoms, and further investigations were conducted by imaging, which clearly demonstrated sacral fractures and which were followed for a period of more than 6 months. Inclusion criteria for this study were: age ≥ 60 years and no history of high energy trauma; patients who presented with clinical symptoms such as tenderness in the sacral area in all three modalities; radiographs, CT, and MRI. The SIF was confirmed by one of the modalities as part of the study. We excluded patients with pelvic malignancy or previous pelvic surgery. Along with the SIF, associated injuries caused, such as fractures of the spine, pelvis, or hip, were considered in this study. Thirty-one osteoporosis, thirty-six osteopenia, and ten normal patients were included in this study.

### 2.2. Image Technique

The investigations included radiograms, CT, and MRI, which were conducted for the patients who were admitted to the hospital with the presenting symptoms. Plain radiographs were basically anteroposterior and lateral radiographs (Figure 1A,B). Plain CT and 3-dimensional (3D) CT images were created using the Aquilion Prime platform (Canon, Tokyo, Japan) (Figure 1C). Imaging conditions were: tube voltage 120 kV; scan speed 0.50 s; slice width 0.5 × 80 mm; and helical pitch 65.0. From this data, the 3D spine image was reconstructed using AIDR 3D enhanced Strong software (Canon). MRI images were taken using Signa HDxt 1.5-T platform (General Electric, Boston, MA, USA).

In Figure 2A,B, T1-weighted sagittal images were taken with a slice thickness 5 mm, slice spacing 1 mm, echo time 8.5 msec, repetition time 400 msec, band width 31.25 Hz/pixel, matrix 256 × 256, field of view 300 mm, and scan time 1:19. T2-weighted sagittal images were taken with a slice thickness 5 mm, slice spacing 1 mm, echo time 110 msec, repetition time 3500 msec, band width 41.67 Hz/pixel, matrix 256 × 256, field of view 300 mm, and scan time 1:09. Fat-suppressed T2-weighted sagittal images were taken with a slice thickness 5 mm, slice spacing 1 mm, echo time 110 msec, repetition time 3500 msec, band width 41.67 Hz/pixel, matrix 256 × 256, field of view 300 mm, and scan time 1:09. T2-weighted coronal images were taken with a slice thickness 3 mm, slice spacing 0.5 mm, echo time 110 msec, repetition time 4000 msec, band width 31.25 Hz/pixel, matrix 256 × 256, field of view 300 mm, and scan time 1:30. T2-weighted axial images were taken with a slice thickness 4 mm, slice spacing 1 mm, echo time 110 msec, repetition time 4000 msec, band width 31.25 Hz/pixel, matrix 256 × 192, field of view 300 mm, and scan time 2:48. T2-weighted fat suppression images were also included.

### 2.3. Evaluation of Images

AO classification [10]; single vertical (unilateral), bilateral vertical, transverse, bilateral with transverse (H); incomplete bilateral with transverse (U); single with transverse (λ/T). Interrater agreement (Kappa coefficients for interrater reliability) in this study was 0.72 and acceptable.

### 2.4. Statistical Analysis

All data were expressed as mean ± standard deviation. Imaging findings were statistically compared between the groups. For comparisons between groups, the Mann-Whitney U test analysis was used to analyze continuous variables, and the chi-squared test was used to analyze dichotomous variables. McNemar’s test has been used for the comparison of the P Values. A *p*-value < 0.05 was defined as statistically significant.

## 3. Results

### 3.1. Sensitivity of Plain Radiograms, CT, and MRI

The sensitivities for SIF were 28.5% in radiographs, 94.2% in CT, and all fractures were detected in MRI, which showed a more complex image pattern of SIF compared with CT (Figure 3). While the radiographs showed a possible fracture with a breach in the cortical continuity and the CT images showed a fracture pattern involving the sacrum, the MRI clearly showed the SIF with associated bony and surrounding soft tissue oedema in T1, T2, and fat suppression images, further confirming its sensitivity, which seemed to be higher than those of CT and plain radiographs (*p* < 0.05, *p* < 0.01). CT sensitivity was higher than that of plain radiograms (*p* < 0.01) (Figure 4).

### 3.2. Accompanying Other Fracture

Single SIFs were 71.4%, 13.0% with other pelvic fractures, 9.1% with other spinal fractures, 3.9% with femoral neck fractures, and others were 3.9%. The most common other fracture was a pelvic fracture (Figure 5).

### 3.3. SIF Fracture Pattern

According to the SIF fracture pattern, the unilateral vertical type was 1.3%, bilateral vertical type was 0%, transverse type was 33.7, H or U type was 40.2%, and λ or T type was 24.7%. The most common type was the H/U type, followed by the transverse and λ types (Figure 6).

### 3.4. Typical SIF Case

Case 1: 87-year-old woman, λ/T-type fracture of SIF (Figure 7).

Case 2: 57-year-old woman, H/U-type fracture of SIF (Figure 8).

Case 3: 89-year-old woman, unilateral vertical-type fracture of SIF (Figure 9).

## 4. Discussion

By definition, a sacral insufficiency fracture is a fracture resulting from physiological stresses on the weakened sacrum, either due to osteoporosis or other factors. Described first by Laurie in 1982 [4], a recent meta-analysis reported 14% of SIF in patients who underwent pelvic radiotherapy for gynecological cancer [14]. The diagnosis could be improved by the increased availability of imaging techniques [15]. Sacral insufficiency fractures must be differentiated from sacral fatigue fractures, though both are included in sacral stress fractures. Pentecoast divided sacral stress fractures into fatigue and insufficiency fractures [16]. Unlike the former, the latter is seen in bones with reduced density that are exposed to trivial/normal trauma. An SIF is an often underdiagnosed condition resulting in significant morbidity in the elderly population. Early diagnosis and treatment are essential to avoid complications of recumbency such as deep vein thrombosis, pulmonary compromise, bed sores, depression, etc. [11]. Most SIFs can be treated conservatively with anti-osteoporosis medicines and analgesics.

Sacral insufficiency fractures are often missed on plain radiographs, with a sensitivity of only 5–35% reported in various studies [9]. Sometimes, even a CT scan can miss the fracture if the cortical integrity is intact. Hence, in such cases, MRI turns out to be superior as a diagnostic tool, even in occult SIF. MRI is helpful not only in the diagnosis of SIF, but also to rule out other associated pathologies such as a missed vertebral fracture or a lumbar canal stenosis [17,18]. Bone scanning is one of the most sensitive techniques for detecting SIF. The H-shaped uptake, which is known as a Honda sign, is sometimes diagnostic for SIF in the proper clinical settings [19]. The classic Honda sign is formed when there are vertical fractures of both sacral alae and a transverse fracture line involving the sacral body [20]. Nowadays, SPECT/CT can be used to diagnose SIF [21]. However, with advancements in recent imaging techniques such as MRI, invasive nuclear radiation imaging methods should only be utilized in limited cases.

There are several classifications for sacral fractures, such as those of Dennis [22] and Roy Camille & Isler [23]. However, these classifications systems are descriptive, and none of these classifications thoroughly describe SIF. According to our experience, we found the AO classification for SIF to be simple and comprehensive [24]. In our classification, the unilateral or bilateral vertical type was mainly caused by a minor fall with a lateral compression of the pelvic ring. The transverse type is caused by a minor fall with a vertical compression of the pelvic ring. H/U type and λ/T are made by a previous combination. There were very few simple vertical fractures in this study because the patients had relatively severe osteoporosis, and thus the sacral fragility could be very severe.

The primary objective of our paper was to compare the sensitivity of the three primary radiological methods, that is, radiographs, CT scan, and MRI. The secondary objective was to identify the ability of CT scan and MRI to correctly classify the fracture pattern. In addition, the incidence of other pelvic and lumbar pathologies on MRI was noted, which might influence the treatment strategy. We found that MRI with fat suppression images was highly sensitive (100%) in detecting SIFs as compared to CT (94.5%) and radiographs (24.2%). This is in accordance with other papers described in the literature. In a study by Cabarrus et al., in their sample size of 67 patients, 75% were detected by CT scan and 100% by MRI [25]. In another study by Henes et al. with a sample size of 38 patients, the sensitivities of CT and MRI were 97% and 100%, respectively [26]. Thus, MRI is a superior diagnostic tool for occult SIF. In a paper by Graul et al. with a sample of 77 subjects, 8% of cases were missed on MRI due to the absence of fat suppression images [10]. This highlights the importance of fat suppression images in detecting SIFs. In addition, our study showed that the commonest variants detected on MRI were the H/U type (40.2%), followed by the transverse (33.7%) and λ/T (24.7%) types. Other associated pathologies identified on MRI were pelvic fractures in 13% of cases and spine fractures in 9.1% of cases. This highlights the importance of MRI in not only identifying SIFs but also associated fractures/pathologies, which were present in 26% of the cases in the present study.

All patients included in the study were treated by conservative management, as per their presentation. The majority of SIFs were reported to respond to conservative treatment [4,27]. The patients who presented with severe symptoms such as a neurological deficit and/or inability to walk opted for surgical management [28,29]. In particular, a strong surgical fixation was necessary if the patients had a previous spinal long fusion [5]. Patients with continued pain but who are poor surgical candidates may benefit from interventional sacroplasty [30,31].

There are several limitations in the present study. This study dose not include a group of patients without SIF. A high variance in gender distribution was noted, as the study group was essentially elderly, and post menopausal women with progressive osteoporosis were a major part of it. A clinical evaluation was not included in the study, as the study criteria were mainly the high specificity in imaging modalities to diagnose the SIF. Our sample size was relatively small, and further studies with larger subjects and a prospective model may be necessary to obtain stronger evidence.

## 5. Conclusions

For sacral insufficiency fractures (SIF), the diagnosis sensitivity of MRI, CT, and radiograms were 100%, 94.2%, and 28.5%, respectively. For the fracture pattern, the incidence of the H/U type was the most common (40.2%), followed by the transverse type (33.7%) and λ/T type (24.7%). The MRI of the lumbar spine including the sacrum with a coronal fat-suppressed T2-weighted image for elderly patients with suddenly increasing low back pain at an early stage is superior to the CT and radiogram. MRI for elderly patients with suddenly increasing low back/sacral pain at an early stage improves early SIF detection, recognition of concomitant pathologies, and adequate treatment for the patients.

## Figures and Tables

**Figure 1 jcm-11-04968-f001:**
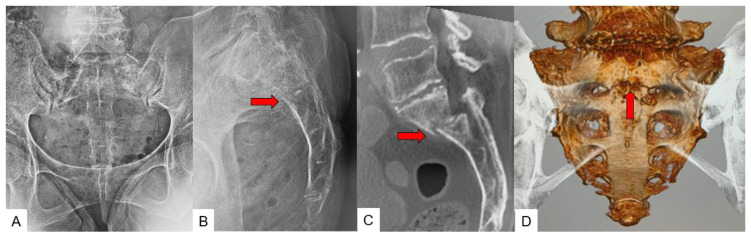
Sacral insufficiency fracture. (**A**): Anteroposterior radiograph, (**B**): Lateral radiograph, (**C**): Sagittal CT, (**D**): 3D CT, red arrows indicate fracture line.

**Figure 2 jcm-11-04968-f002:**
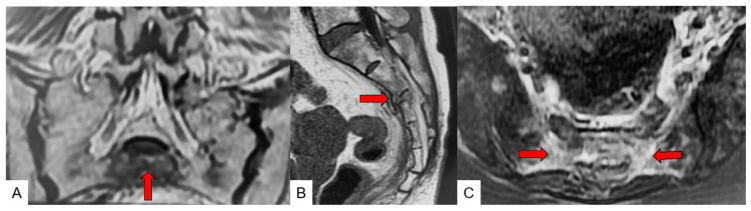
Sacral insufficiency fracture. (**A**): Coronal T2-weighted MR image, (**B**): Sagittal T1-weighted MR image, (**C**): Axial T2-weighted MR image, red arrows indicate fracture line.

**Figure 3 jcm-11-04968-f003:**
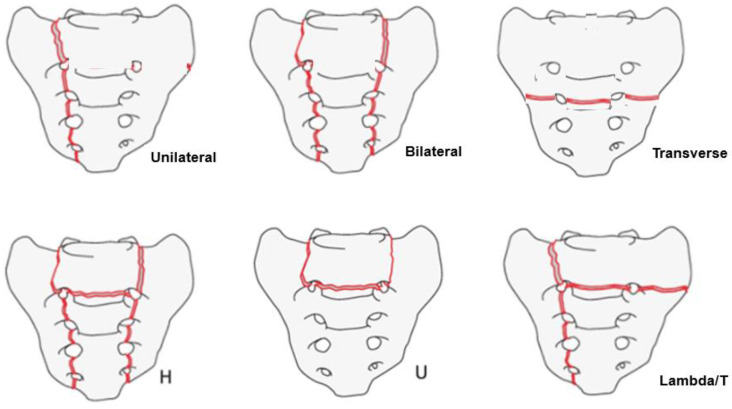
Fracture pattern of SIF.

**Figure 4 jcm-11-04968-f004:**
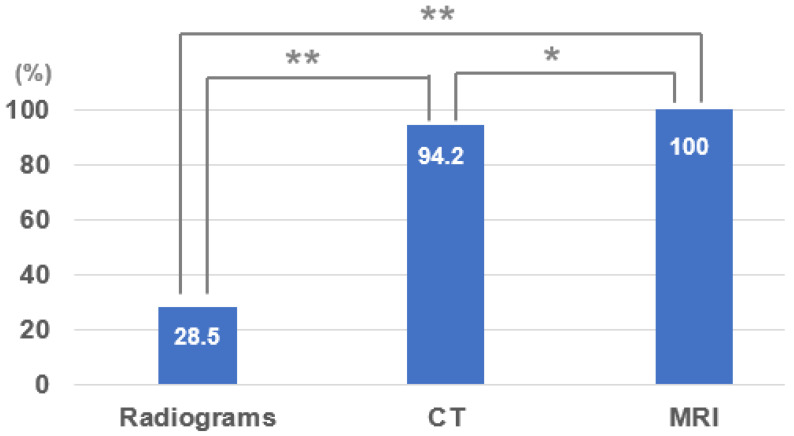
SIF diagnosis sensitivity of three modalities; * *p* < 0.05, ** *p* < 0.01. CT: computed tomography, MRI: magnetic resonance imaging.

**Figure 5 jcm-11-04968-f005:**
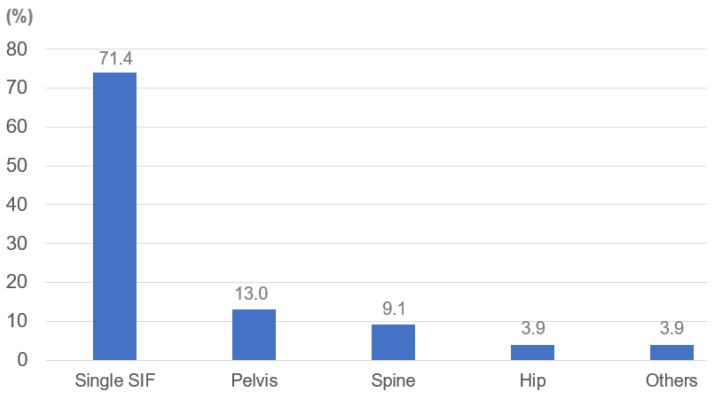
Other fracture.

**Figure 6 jcm-11-04968-f006:**
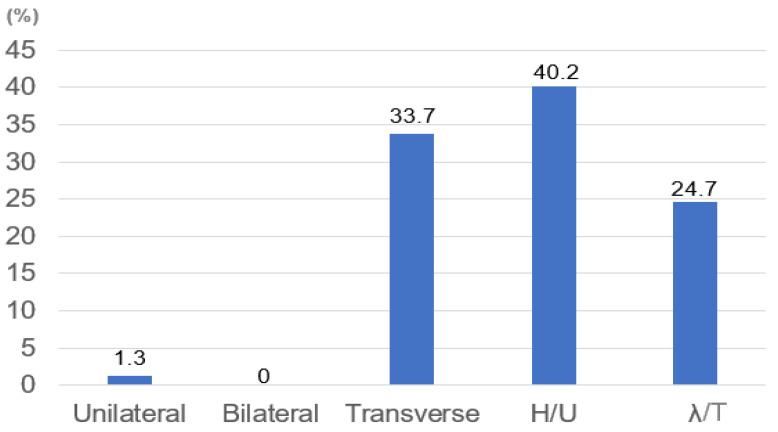
SIF fracture pattern.

**Figure 7 jcm-11-04968-f007:**
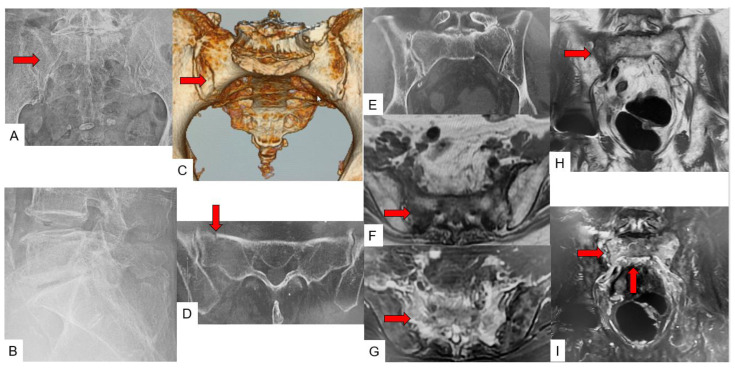
87-year-old woman, λ/T-type fracture of SIF. (**A**): Anteroposterior radiograph, (**B**): Lateral radiograph, (**C**): 3D CT, (**D**): Axial CT, (**E**): Coronal CT, (**F**): Axial T1-weighted MR image, (**G**): Axial fat-suppressed T2-weighted MR image, (**H**): Coronal T1-weighted MR image, (**I**): Coronal fat-suppressed T2-weighted MR image, red arrows indicate fracture line.

**Figure 8 jcm-11-04968-f008:**
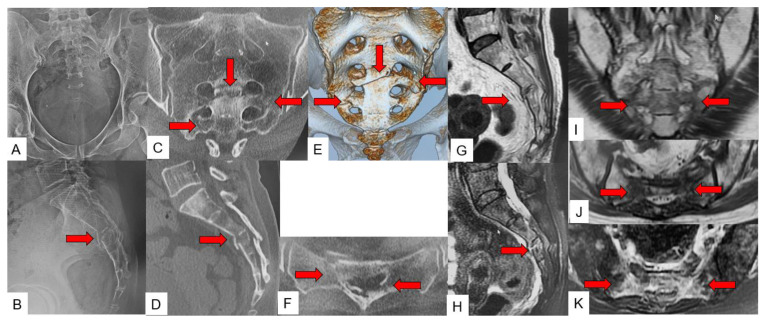
57-year-old woman, H/U-type fracture of SIF. (**A**): Anteroposterior radiograph, (**B**): Lateral radiograph, (**C**): Coronal CT, (**D**): Sagittal CT, (**E**): 3D CT, (**F**): Axial CT, (**G**): Sagittal T1-weighted MR image, (**H**): Sagittal fat-suppressed T2-weighted MR image, (**I**): Coronal T1-weighted MR image, (**J**): Axial T1-weighted MR image, (**K**): Axial fat-suppressed T2-weighted MR image, red arrows indicate fracture line.

**Figure 9 jcm-11-04968-f009:**
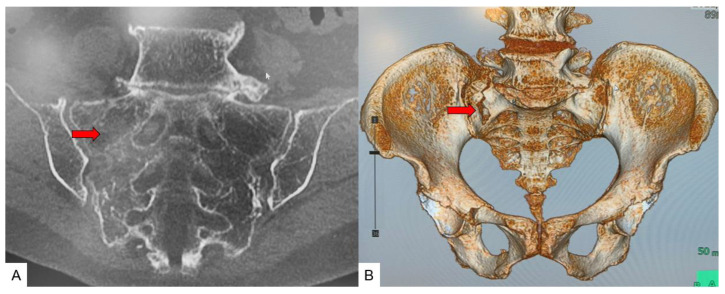
89-year-old woman, unilateral vertical-type fracture of SIF. (**A**): Coronal CT, (**B**): 3D CT, red arrows indicate fracture line.

**Table 1 jcm-11-04968-t001:** Patient demographics.

	N = 77
Gender (Men:Women)	11:66
Age (mean ± S.D.) (year)	80.3 ± 10.4
Height (mean ± S.D.) (cm)	150.6 ± 8.6
Body weight (mean ± S.D.) (kg)	47.4 ± 9.2
Body mass index (mean ± S.D.) (kg/m^2^)	21.0 ± 3.9
Bone mineral density lumbar (g/cm^2^) T-score lumbar	0.747 ± 0.16 −2.4 ± 0.5
Bone mineral density hip (g/cm^2^) T-score hip	0.583 ± 0.10 −2.3 ± 1.5

## Data Availability

The data presented in this study are available in the article.

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
