# Peer review of "Superiority of MRI for Evaluation of Sacral Insufficiency Fracture"

_jcm, 2022, doi:10.3390/jcm11174968_

Round 1

Reviewer 1 Report

Important topic, but there are aspects of the methods that may need some  clarification.

Introduction - the author included a result in the introduction section (line 46), consider moving the ‘mean age 80.03’ to the results section.

Methods - the authors mention that the sensitivity was calculated (defined as true positive divided by the sum of true positive and false negative), but does not specify what was the standard test (golden standard) used to obtain the sensitivity (sensitivity and specificity is calculated in relation to a standard [golden] test). It seems that the MRI was used as the standard, which obviously resulted in 100% sensitivity for this method (when compared to itself). Possibly another design would be more indicated to evaluate that, since the MRI was used as the index and also standard test (redundancy).

Methods - Due to the extrinsic relation between sensitivity and specificity, it would be reasonable to provide both sensitivity and specificity estimates. Again, due to the use of MRI as the index and standard test, it’s likely that both values would be 100% for this method (MRI) using a redundant analysis.

Methods - authors mentioned that it is a retrospective observational study (abstract line 14). They also mention that an informed consent was obtained from each patient prior to participation in the study (methods line 54). Was this consent a regular consent for the performing the imaging exam or a specific consent to have the data analyzed in this retrospective study ? Many IRBs don’t request a new consent for retrospective studies. Please, clarify.

Methods - It is said that the study included patients from June 2018 to January 2022 (methods, line 55). It is also said that the patients were followed for ‘a period of more than one year’. Since we are in August 2022, it may be more correct to say that after at least January 2023.

Results - not clear if all patients (n=77) underwent the three exams (x-ray, CT, and MRI). Should provide this information in the text, a table, or in the Figure 4. Per possible study design (assuming MRI was the standard) it is presumed that all patients underwent MRI, but if not all 77 patients underwent the three exams, the number of patients that underwent the pairs x-ray/MRI and CT/MRI should be provided.

Results - the authors indicate in the abstract that:

“All patients except one were treated with conservative treatment.” —> abstract lines 23-24

But in the results it is mentioned:

“All patients were managed with conservative treatment.” —> results line 112

It seems that one patient treated non-conservatively appeared in the abstract but was not mentioned in the results.

Since the study focus on the radiological description of the fractures and attempt to estimate sensitivity of different modalities for detection, it may be more prudent to not mention treatment, specially if the data was not collected with this focus.

Discussion - the authors start the discussion mentioning that there is little data on the incidence of these fractures. However, a recent meta-analysis of almost 4000 patients estimated hat ~14% of the patients that underwent pelvic radiotherapy for gynecological cancer present with this complication (PMID: 32580930). This relative high incidence level highlights the importance of the topic, specially after pelvic RT for gynecological cancer.

Author Response

To Respective Reviewer 1

We appreciate your valuable and thoughtful comments.

Important topic, but there are aspects of the methods that may need some  clarification.

Introduction - the author included a result in the introduction section (line 46), consider moving the ‘mean age 80.03’ to the results section.

Thank you for your important comment. We moved that to the result section.

Methods - the authors mention that the sensitivity was calculated (defined as true positive divided by the sum of true positive and false negative), but does not specify what was the standard test (golden standard) used to obtain the sensitivity (sensitivity and specificity is calculated in relation to a standard [golden] test). It seems that the MRI was used as the standard, which obviously resulted in 100% sensitivity for this method (when compared to itself). Possibly another design would be more indicated to evaluate that, since the MRI was used as the index and also standard test (redundancy).

We appreciate your valuable comment. The standard test is not MRI. Patients with sacral insufficiency fracture (SIF) complain severe sacral pain and gait difficulty due to pain. We always take radiograms and MRI, follow these patients very carefully and usually recommend to be admitted to the hospital. We think CT/3D-CT is very useful to evaluate fracture line, so we also take CT almost all cases. The standard test is clinical symptoms and confirmed the fracture of the patient at least one modality at the follow-up.  

Methods - Due to the extrinsic relation between sensitivity and specificity, it would be reasonable to provide both sensitivity and specificity estimates. Again, due to the use of MRI as the index and standard test, it’s likely that both values would be 100% for this method (MRI) using a redundant analysis.

Thank you for your thoughtful comments. As you mentioned, MRI can detect no only fracture but also bone burse/ edema/ inflammatory conditions. In this retrospective study, we evaluate only true SIF patients. We perform a lot of MRIs for low back pain patients. Please understand that it is very difficult to calculate specificity.

Methods - authors mentioned that it is a retrospective observational study (abstract line 14). They also mention that an informed consent was obtained from each patient prior to participation in the study (methods line 54). Was this consent a regular consent for the performing the imaging exam or a specific consent to have the data analyzed in this retrospective study ? Many IRBs don’t request a new consent for retrospective studies. Please, clarify.

We’ d like to thank you for your important comment.

YES. In our hospital, we always ask a comprehensive consent, of which the patient can refuse,  for out-patients. Our IRBs don’t request a new consent for retrospective studies.

Methods - It is said that the study included patients from June 2018 to January 2022 (methods, line 55). It is also said that the patients were followed for ‘a period of more than one year’. Since we are in August 2022, it may be more correct to say that after at least January 2023.

Thank you for your comment. We changed the sentence according to your advice.

“a period of more than 6 months

Results - not clear if all patients (n=77) underwent the three exams (x-ray, CT, and MRI). Should provide this information in the text, a table, or in the Figure 4. Per possible study design (assuming MRI was the standard) it is presumed that all patients underwent MRI, but if not all 77 patients underwent the three exams, the number of patients that underwent the pairs x-ray/MRI and CT/MRI should be provided.

We appreciate your important comments. We usually perform three modalities.

The sentence is inserted in inclusion criteria.

Results - the authors indicate in the abstract that:

“All patients except one were treated with conservative treatment.” —> abstract lines 23-24

But in the results it is mentioned:

“All patients were managed with conservative treatment.” —> results line 112

It seems that one patient treated non-conservatively appeared in the abstract but was not mentioned in the results.

Since the study focus on the radiological description of the fractures and attempt to estimate sensitivity of different modalities for detection, it may be more prudent to not mention treatment, specially if the data was not collected with this focus.

Thank you for your helpful comment. We deleted the clinical part.

Discussion - the authors start the discussion mentioning that there is little data on the incidence of these fractures. However, a recent meta-analysis of almost 4000 patients estimated hat ~14% of the patients that underwent pelvic radiotherapy for gynecological cancer present with this complication (PMID: 32580930). This relative high incidence level highlights the importance of the topic, specially after pelvic RT for gynecological cancer.

We appreciate your valuable comments. We changed the sentence as follows;

a recent meta-analysis reported 14% of SIF in patients who underwent pelvic radiotherapy for gynecological cancer [A].

A: . Chung YK, Lee YK, Yoon BH, Suh DH, Koo KH. Pelvic Insufficiency Fractures in Cervical Cancer After Radiation Therapy: A Meta-Analysis and Review. In Vivo. 2021;35(2):1109-1115.

Reviewer 2 Report

The present paper clearly compares several strategies to evaluate sacral insufficiency fractures (SIF), such as radiographs, CT scan and MRI. Additionally, the work tries to find the best imaging techniques to classify the fracture pattern.

Even if the context is of particular scientific interest and the methodological approach is well described, two issues arise:

1. The introduction lacks in presenting the concrete problem linked to SIF. Which is the impact on worldwide population? Which are the current diagnostical tools? An in-depth review of the state of the art is missing. Additionally, since bone fractures occur at the multi-scale, which are the current strategies for their early detection? See https://doi.org/10.3390/ma14051240 , https://doi.org/10.1002/adhm.201500070

I strongly suggest to add a paragraph in the introduction with a clear mention to the complexity of bone fracture detection.

2.The paper does not quantify the SIF extension in the three different imaging techniques. Since the extension of a fracture is a core aspect to determine its severity, I suggest to improve the manuscript by quantifying the SIF extension from the clinical images. This would result in an additional indictor of imaging technique sensitivity.

Other comments are the following:

3. Line 47: provide a reference

4. Line 157-158:  why do the authors write “The incidence is increasing because of availability of imaging techniques”? This is not correct, the diagnosis could be improved by the increased availability of imaging techniques.

5. The conclusion should summarize the main author findings.

Author Response

To Respective Reviewer 2

We appreciate your valuable and thoughtful comments.

The present paper clearly compares several strategies to evaluate sacral insufficiency fractures (SIF), such as radiographs, CT scan and MRI. Additionally, the work tries to find the best imaging techniques to classify the fracture pattern.

Even if the context is of particular scientific interest and the methodological approach is well described, two issues arise:

  1. The introduction lacks in presenting the concrete problem linked to SIF. Which is the impact on worldwide population? Which are the current diagnostical tools? An in-depth review of the state of the art is missing. Additionally, since bone fractures occur at the multi-scale, which are the current strategies for their early detection? See https://doi.org/10.3390/ma14051240 , https://doi.org/10.1002/adhm.201500070

I strongly suggest to add a paragraph in the introduction with a clear mention to the complexity of bone fracture detection.

Thank you for your important comments. We added the sentences as follows.

Human bone becomes more prone to fracture and an incidence of osteoporotic or insufficient fracture increase with age. Age-related fracture is an increasing health and economic concern and increased 50% in twenty years in USA [B]. In Asia, more than 50% worldwide osteoporotic fracturs will occur by 2050 [C]. In Europe, the cost of osteoporosis will become 37 billion Euro [C]. In order to resize the impact of age-related bone fractures and to start reducing their health and economic influence, early diagnosis is key [D].

B: Burge, R.; Dawson-Hughes, B.; Solomon, D.H.; Wong, J.B.; King, A.; Tosteson, A. Incidence and Economic Burden of OsteoporosisRelated Fractures in the United States, 2005–2025. J. Bone Miner. Res. 2007, 22, 465–475

C: Tatangelo, G.; Watts, J.; Lim, K.; Connaughton, C.; Abimanyi-Ochom, J.; Borgström, F.; Nicholson, G.C.; Shore-Lorenti, C.; Stuart, A.L.; Iuliano-Burns, S.; et al. The Cost of Osteoporosis, Osteopenia, and Associated Fractures in Australia in 2017. J. Bone Miner. Res. 2019, 34, 616–625.

D: Buccino F, Colombo C, Vergani LM. A Review on Multiscale Bone Damage: From the Clinical to the Research Perspective. Materials (Basel). 2021 Mar 5;14(5):1240.

  1. The paper does not quantify the SIF extension in the three different imaging techniques. Since the extension of a fracture is a core aspect to determine its severity, I suggest to improve the manuscript by quantifying the SIF extension from the clinical images. This would result in an additional indictor of imaging technique sensitivity.

We appreciate your valuable comments. We are awfully sorry that it is very difficult to e quantify the SIF extension using MRI or CT. As we mentioned, the primary objective of this study was to evaluate sensitivity in SIF according to plain radiographs, CT, and MRI. The second objective was to identify SIF fracture patterns by our classification.

Other comments are the following:

  1. Line 47: provide a reference

Thank you for your comment. We added the reference.

E: Demographic statistics in Japan 2021, Ministry of Health, Labour and Welfare, https://www.mhlw.go.jp/toukei/saikin/hw/jinkou/geppo/nengai21/index.html

  1. Line 157-158:  why do the authors write “The incidence is increasing because of availability of imaging techniques”? This is not correct, the diagnosis could be improved by the increased availability of imaging techniques.

We appreciate your thoughtful comments. We changed the sentences as you mentioned.

The diagnosis could be improved by the increased availability of imaging techniques [11].

  1. The conclusion should summarize the main author findings.

Thank you for your comment. We changed the conclusion as follows;

For sacral insufficiency fracture (SIF), the diagnosis sensitivity of MRI, CT and radiograms were 100%, 94.2%, and 28.5%, respectively. For the fracture pattern, the incidence of H/U type was the most common (40.2%), followed by transverse type (33.7%) and λ/T type (24.7%). of the lumbar spine including the sacrum with coronal fat-supprested T2-weighted image for elderly patients with suddenly increasing low back pain at an early stage is superior to CT and radiogram. MRI for elderly patients with suddenly increasing low back/ sacral pain at an early stage improves early SIF detection, recognition of concomitant pathologies, and adequate treatment for the patients.

Round 2

Reviewer 1 Report

In a study evaluating a measure of radiological accuracy, 'standard test' refers to the standard method used for calculation of sensitivity of the index test. It is not referred to the 'standard' clinical criteria used for diagnosis in clinical practice.

Consider explaining in methods how the sensitivity (main outcome) was calculated in the present study to obtain the result in lines 118-119:

"The sensitivities for SIF were 28.5% in Radiographs, 94.2% in CT, and all fractures 118 were detected in MRI (...)"